# Polybrominated Diphenyl Ether Serum Concentrations and Depressive Symptomatology in Pregnant African American Women

**DOI:** 10.3390/ijerph18073614

**Published:** 2021-03-31

**Authors:** Abby D. Mutic, Dana Boyd Barr, Vicki S. Hertzberg, Patricia A. Brennan, Anne L. Dunlop, Linda A. McCauley

**Affiliations:** 1Nell Hodgson Woodruff School of Nursing, Emory University, Atlanta, GA 30322, USA; vhertzb@emory.edu (V.S.H.); amlang@emory.edu (A.L.D.); linda.mccauley@emory.edu (L.A.M.); 2Gangarosa Department of Environmental Health, Rollins School of Public Health, Emory University, Atlanta, GA 30322, USA; dbbarr@emory.edu; 3Department of Psychology, Emory University, Atlanta, GA 30322, USA; pbren01@emory.edu

**Keywords:** polybrominated diphenyl ether, antenatal depression, endocrine disrupting chemical, neuroendocrine

## Abstract

(1) Polybrominated diphenyl ethers (PBDEs) were widely produced in the United States until 2004 but remain highly persistent in the environment. The potential for PBDEs to disrupt normal neuroendocrine pathways resulting in depression and other neurological symptoms is largely understudied. This study examined whether PBDE exposure in pregnant women was associated with antenatal depressive symptomatology. (2) Data were collected from 193 African American pregnant women at 8–14 weeks gestation. Serum PBDEs and depressive symptoms were analyzed and a mixture effect was calculated. (3) Urban pregnant African American women in the Southeastern United States had a high risk of depression (27%) compared to the National average. Increased levels of PBDEs were found. BDE-47 and -99 exposures are significantly associated with depressive symptomatology in the pregnant cohort. The weighted body burden estimate of the PBDE mixture was associated with a higher risk of mild to moderate depression using an Edinburgh Depression Scale cutoff score of ≥10 (OR = 2.93; CI 1.18, 7.82). (4) Since antenatal depression may worsen in postpartum, reducing PBDE exposure may have significant clinical implications.

## 1. Introduction

Polybrominated diphenyl ethers (PBDEs) are a family of chemicals with a common structure of a diphenyl ether molecule attached to one to ten bromine atoms [1]. PBDEs were once added to consumer products and materials to attenuate the risk of fire and to increase the escape time in the event of a fire. A voluntary phase-out of pentaBDEs and octaBDEs began in the United States in 2004 and later, decaBDEs; however, they remain prevalent in the environment over 10 years later. PBDE chemicals are not covalently bound to materials and leach from consumer products into the environment [2]. Additionally, they are not easily biodegraded, and thus, remain persistent and ubiquitous in the environment. Levels of PBDEs in humans and the environment are higher in North America than in other regions of the world, likely because of the widespread commercial use of PBDE mixtures in the United States: decaBDE and pentaBDE (widely used in textiles, plastics, electronics, and polyurethane foam) [2]. The chemicals are highly detectable in air, dust, sediment, soil, and ground water [2] and, with their hydrophobic properties, PBDEs can bioaccumulate in lipid-rich human tissue [3]. Exposure to PBDEs has been associated with thyroid disease [4,5,6], reproductive changes [7], neurodevelopmental deficits [8,9,10,11,12,13], and gestational diabetes [14]. The potential for environmental toxicants such as PBDEs to disrupt normal neuroendocrine pathways resulting in depression and other neurological symptoms has been largely unstudied. Literature regarding PBDE exposure and subsequent short- and long-term health effects has focused on neurological effects among infants and children. To date, only two studies have investigated the effects of PBDE exposure outside of pregnancy and subsequent neuropsychological effects in the woman [15,16]. In this study, we focus on PBDE exposure in pregnant women ≥ 18 years old and their association with depressive symptoms.

Clear connections between PBDE levels and thyroid disruption have been made [4,5,17,18,19,20]. PBDEs have similar shape and structure to thyroid hormones and are thought to mimic thyroid activities as a result. Furthermore, the essentiality of thyroid homeostasis for cognition and mood stability has been long recognized [21,22,23,24]. Thyroid disease can present with depressive symptoms, loss of concentration, and other neurological conditions when thyroid stimulating hormone is either up regulated or downregulated [25]. Of the limited data specific to PBDE exposure and risk of depression within the same individual, results vary [15,16,26]. Vuong et al. (2020) measured PBDEs in maternal serum and depressive symptoms among pregnant women in their second trimester [26]. PBDE congeners −28, −47, −100, −153, and ∑PBDEs were significantly associated with depression scores [26]. In other studies, neuropsychological measures such as hostility, aggression, and temperament among children or learning and memory in older adults were more clearly associated with PBDE concentrations [15,16]. Additional neurological associations have been reported among mixtures of polychlorinated biphenyl (PCB) and PBDE exposures [16]. PCBs and PBDEs have been postulated to share mechanistic actions and toxic effects [27,28,29,30] and others endocrine disrupting chemicals and have been correlated with depression-like symptoms as well [16,31,32,33,34].

Pregnant women may be an overlooked vulnerable population to the effects of toxicants due to normal physiological changes in pregnancy such as higher lipid profiles and thyroid strain [35,36,37]. African Americans specifically may also be at elevated risk of toxicant exposure. Evidence of excess toxicant exposure among African Americans compared to Caucasians has been noted during pregnancy and associated with several adverse birth outcomes such as reduced birth length, weight, and head circumference, and spontaneous abortion. According to the most recent 2003–2008 National Health and Nutrition Examination Survey (NHANES) population data, non-Hispanic Blacks have elevated PBDE body burden when compared to non-Hispanic Whites [38,39]. Cross-sectional studies have found similar findings predicting non-Hispanic Blacks are nine times more likely to have high concentrations of persistent organic pollutants (which include PBDEs) than non-Hispanic Whites after adjusting for income [40].

African American women may also be at increased risk for perinatal depression- a mood disorder present in the antenatal or postpartum period for up to one year. Melville et al. [41] estimated nearly 19% of African American women met diagnostic criteria for either minor or major depression during pregnancy, considerably higher than the national average of 8.5% [42]. African American women of low income have the highest reported prevalence of perinatal depression (28%) in the U.S. [43]. Major depressive disorder is associated with adverse outcomes in many aspects of role performance [44]. Specifically, major depression is associated with poor outcomes in marital or relationship functioning [45], parental functioning [46], absenteeism and low work performance [47,48], and personal income or household earnings [49]. Unfortunately, some populations of African Americans are less likely to be diagnosed and treated for depressive symptoms compared to White patients [50], and across subgroups, African Americans are more likely to have persistent disorders after becoming ill [51]. Additionally, mental health concerns are often overlooked, misdiagnosed, mistreated, or not treated and, historically, both patients and providers have poor follow-up of mental health concerns.

Few quantitative studies have examined health effects of PBDE exposures in pregnant women and, to our knowledge, none have focused on African American women of varying socioeconomic status. The study was based in the urban Southeast where, despite cohort distinctions existing in housing, climate, traffic, diet, culture, and racial/ethnic composition, no cohort studies have characterized their environmental exposures and related maternal health outcomes. We hypothesized a positive relationship between PBDE serum concentrations and depression among pregnant African American women. The objective of this study was to investigate whether PBDE exposure in pregnant African American women was associated with depressive symptomatology during pregnancy as individual and combined effects of exposure.

## 2. Materials and Methods

### 2.1. Recruitment of Study Sample

This study used data drawn from a larger longitudinal pregnancy and birth cohort of 487 women aimed to investigate the maternal prenatal microbiome as a predictor of birth outcomes [52]. Women were enrolled in the study between 2014 and 2017 and were included if they self-identified as African American, had a singleton pregnancy, were 18–35 years of age, resided in an urban Georgia area, and could comprehend and speak English. None of the participants had a history or current report of thyroid or parathyroid disease, depression, or diabetes. Women were recruited during their first trimester from prenatal clinics serving two local hospitals. Members of this SES-diverse cohort were individually informed of the study protocol and written consent was obtained prior to participation. All study procedures were approved on 10 September 2013 and conducted in accordance with the Institutional Review Board at Emory University (IRB00068441). Participants were followed through delivery. For this analysis, a subset of 193 women were included if they had serum PBDE concentrations and complete Edinburgh Depression Scale scores (Appendix A). A comparison between the cohort enrollees included in this analysis and those who were not indicates the groups are equally representative of the larger cohort of 487 women (Appendix A). Measures (demographic, clinical, and PBDE data) were collected in person at the initial prenatal clinic visit by a research team member via paper questionnaires, electronic medical record documentation, and maternal venous blood draw. All study procedures were approved and reviewed by the Institutional Review Board at Emory University.

### 2.2. Demographic Data

Socio-demographic variables included age, relationship status (not in a relationship, in a relationship not cohabitating, and in a relationship and cohabitating), years of education, and insurance status. Relationship status was used instead of being married since our sample was largely unmarried. The poverty/income ratio (PIR) variables were determined from family size and household income data per the method of the 2013 U.S. Census Bureau [53]. PIR values below 1.00 are below the official poverty threshold. This study categorized participants as qualifying for Medicaid insurance with PIR <100%, Medicaid insurance with PIR ≤200%, and private insurance.

### 2.3. Clinical Data

Health data extracted from electronic medical records included height, weight, parity, past and current tobacco use, and alcohol or drug use in the first trimester of pregnancy. Pregnancy dating was based on the participant’s known last menstrual period or first trimester ultrasound if a discrepancy of ± 5–7 days exists per standard dating protocol [52]. Depression data was collected by trained research coordinators at 8–14 weeks gestation. Depression was defined using the Edinburgh Depression Scale (EDS); a 10-item scale of depressive symptoms experienced in the last seven days [54]. Originally designed to predict mild to moderate depression in the postpartum period, the EDS has been validated and vetted as an effective screening tool for antenatal women [55,56] especially given its significant association with postnatal depression [57]. The EDS is a reliable tool to predict antenatal combined major or minor depression [58] with psychometric properties of 64–87% sensitivity, 73–96% specificity, and 73% positive predictive value [56]. Individual question scores were summed, and total scores ranged from 0–30 with higher scores associated with a higher risk of depression.

The EDS has been validated in multiple race and ethnic groups [54,56], with varied cut-off prediction scores based on socioeconomic status, race [42], and gender differences [59]. Additionally, researchers have argued that cultural differences and timing of screenings influence clinical cut-off scores [42,60,61,62,63]. To this end, four cut-off scores were explored (≥7, ≥8, ≥9, and the universal standard ≥10) to determine the optimal cutoff value to detect depressive symptomatology in our African American study sample [54]. Receiver operating characteristic curves were constructed to visualize and compare the specificity of the cutoff scores. An optimal EDS cutoff of ≥10, area under the curve 0.6867, was determined to differentiate the groups of high versus low or no risk of depression (Figure 1).

### 2.4. Polybrominated Diphenyl Ether

Concentrations of PBDE congeners −47, −85, −99, −100, −153, and −154 were measured in maternal serum using a modification of a previous method [64,65]. Samples were fortified with isotopically labeled analogues of the target chemicals, homogenized and deprotonated. Supernatants were extracted twice with hexane and dichloromethane and passed through an activated silica gel column to remove residual biogenic material. Sample extracts were concentrated and analyzed using gas chromatography-tandem mass spectrometry with isotope dilution calibration. The limits of detection (LODs) were in the low pg/mL range (Appendix A). Values below the LOD were replaced with a value equal to the LOD, divided by the square root of two [66]. Maternal serum total cholesterol and free triglycerides were not collected, and therefore lipid adjustment was not done.

### 2.5. Covariates and Potential Confounders

To test the relationship between PBDE congeners and depression, covariates and/or potential confounders were considered a priori and reassessed after bivariate analyses. The confounders identified a priori were age, gravidity, relationship status, education, body mass index (BMI), and income based on associations noted in previous literature [67,68,69,70]. Income had >20% missing values (n = 39), so insurance type was used as a proxy. Parity, indicating the number of pregnancies reaching viable gestational age, was tested as a covariate. During bivariate analysis, age, relationship status, education, and parity were dropped because no clear differences in the depression groups were observed.

### 2.6. Statistical Analysis

PBDE congeners were log transformed (base 10) to reduce the influence of outliers and to conform to normality. For congeners with a 50% or greater presence (−47, −99, and −100), geometric means were used to describe PBDE data and reduce the right skewed data distribution resulting from imputed data. Congeners −85 and −154 were not detected in any samples (above instrumental LOD) and were therefore not reported and excluded from analysis. BDE-153 was detected in only 14% of the samples and was subsequently not used in further analyses.

We examined univariate associations within demographic characteristics, PBDE concentrations, and EDS scores. Geometric means were used to describe PBDE concentrations and to compare to 2013–2014 NHANES data restricted to non-Hispanic Black women aged 18–35 years. Wilcoxon nonparametric tests were used to compare PBDE congeners between low and high depression groups. Parametric tests of association were completed for covariates and depression variables. Spearman rank-order and Kruskal-Wallis tests were conducted to compare means among skewed data. Bivariate analysis between BDE-47 and depression groups yielded unequal variances so Satterthwaite test statistic was reported. Missing data were explored within each covariate and accounted for <10% of the sample and were not further manipulated.

Each PBDE congener was regressed with confounding variables in linear and logistic regression models to investigate associations with depression. Exposure to a PBDE mixture (congeners −47, −99, and −100) was evaluated using a weighted quantile sum (WQS) approach in conjunction with multiple linear and nonlinear logistic regression [71]. The WQS method estimates a weighted linear index corresponding to chosen quantiles of PBDEs. Bootstrap sampling is used to empirically determine the weights, constrained between 0 and 1 and summed to 1. Since environmental exposures co-occur, interact, and are highly correlated, the WQS method reduces dimensionality and addresses multicollinearity [72]. The WQS index was placed in the best-fit model where exp (β1) is the odds ratio associated with a unit (quartile) increase in the weighted quartile sum of PBDE exposures. The WQS index was regressed using an EDS cutoff ≥10. The significance of the test represents a test for a mixture effect. Statistical significance was set at 0.05 and all analyses were performed using SAS v9.4^®^ (SAS Institute Inc., Cary, NC, USA) and R v3.4 software (R Foundation for Statistical Computing, Vienna, Austria). 

## 3. Results

### 3.1. Sample Characteristics

Basic descriptive statistics for the total sample of 193 women are presented in Table 1 and Table 2. Of the total sample, 52 women (26.9%) were categorized as having a high risk of depression (EDS cutoff ≥10) and 141 (73.1%) had a low or no risk of depression. In this sample, the alpha coefficient was 0.85. The majority of participants were in a relationship and cohabitating (53.4%), had at least some college education (48.2%), and qualified for Medicaid health insurance (77.8%) (Table 1). Both high and low or no risk of depression groups had an approximately normal distribution of educational attainment. Approximately half were overweight or obese (54.4%) and nulliparous (48.2%). A small proportion reported consuming alcohol (4.2%) or marijuana (17.1%) in the last month, or ever smoked tobacco (10.9%) (Table 2). The women’s mean BMI scores were significantly different between the depression groups. Substance use in the last month and ever used tobacco were also associated with having an EDS score of ≥10.

### 3.2. Polybrominated Diphenyl Ethers

When comparing the 2013–2014 NHANES data reflecting the pooled samples of non-Hispanic Black women aged 20–39, our study population had higher levels of BDE-47 and −99 but lower levels of BDE-100 (Table 3). Consistent with previous literature, PBDE congeners were significantly intercorrelated (r = 0.6–0.7, *p* < 0.001), limiting statistical methods. Although not strong, serum BDE-47 concentrations and marijuana use were positively associated (r = 0.12, *p* = 0.03). Neither tobacco nor alcohol use were associated with any congener.

### 3.3. Polybrominated Diphenyl Ethers and Depression Risk

Wilcoxon signed-ranks tests revealed levels of BDE-47 and -99 were significantly different between low and high depression groups (z = 2.1014, *p* = 0.04 and z = 2.2881, *p* = 0.02, respectively) (Table 4). As concentrations of BDE-47 in the serum increase, the probability of having depressive symptoms also increases (Figure 2). However, this association only accounted for 6% of the total variability.

All covariates and confounding variables identified a priori and confirmed with bivariate analyses were entered into the multivariate model. The best-fit model controlled for relationship status, insurance type, and BMI to test for associations between PBDEs and depression.

In the adjusted regression analysis, models were used to examine the associations between PBDEs (Table 5), and depressive symptoms. Statistical significance was detected with BDE-47 and -99, but not BDE-100. For every one unit increase in concentration, the risk of being mild to moderately depressed increased by a factor of 4.43 for BDE-47, 1.58 for BDE-99, and 1.23 for BDE-100 adjusting for relationship status, insurance type, and BMI. No issues of multicollinearity existed since all various inflation factors values were low (<2). The Hosmer-Lemeshow goodness of fit (0.177–0.726) suggests good fit for each model.

To show which PBDE congener had the most robust association with depressive symptoms, a PBDE WQS index was regressed with the dichotomous depression variable and presented in Table 6. The distribution of weights comprising the index was led by BDE-47 (w = 0.71), followed by BDE-100 (w = 0.29), then BDE-99 (w = 0. 00000000817). The fitted coefficients exp (β) provided an estimated odds ratio for risk of depression resulting from the PBDE mixture. Specifically, increases in the weighted PBDE index were significantly associated with higher depressive symptoms after adjusting for relationship status, insurance type, and BMI (OR = 2.93; CI 1.18, 7.82).

## 4. Discussion

The purpose of this study was to examine the relationship between concentrations of PBDEs in serum and depressive symptoms in pregnant African American women, a vulnerable and traditionally understudied population in the context of both PBDE exposure and prenatal mental health outcomes. The bulk of PBDE and health outcome research has centered on maternal exposures and fetal developmental outcomes but, as concentrations continue to be detected remote from the production phase-out, health concerns beyond early childhood should be explored. Cowell et al. [73] found no significant differences in PBDE cord blood concentrations collected before and after the phase out supporting the persistent properties and ongoing release of the chemicals from products. Exposures and their body burdens are not well-characterized in pregnant women versus nonpregnant. In our sample of 193 pregnant women with no known occupational or excessive PBDE exposure, 100% had detectable concentrations of BDE-47 in their blood. Blood concentrations from this study are similar to serum concentrations from the most recent 2013–2014 NHANES national sample of non-Hispanic Black women of reproductive ages. Findings indicate positive associations between PBDE congeners, the PBDE mixture, and high depressive symptoms among our study population. The weighted PBDE index identified BDE-47 to be the “bad actor” within the mixture labeling it as the most influential congener. We anticipated this finding, since it is highly prevalent across multiple matrixes [74,75,76]. Due to the small amount of variability in the final regression models, further studies should focus on potential covariates that could better explain the relationship between serum PBDE concentration and prevalence of depressive symptoms such as racial trauma and discrimination, domestic violence, race-related barriers in healthcare, poverty or unemployment [77,78,79,80,81]. We understand that humans are exposed to many environmental chemicals at once, and the preliminary findings in this study using the WQS method broadens our perspectives on analyzing highly correlated variables and provides better estimates of the PBDE mixture effect. For more comprehensive mixture estimates, additional endocrine disrupting chemicals that may disproportionately affect African Americans should be considered in future analyses. Theoretically, endocrine disruptors have similar health effects and may be working through similar pathways to affect mood [82,83]. Assessing for exposure to other endocrine disruptors and neuropsychological outcomes among African American women could provide insight into shared metabolic pathways.

Consistent with previous studies of tobacco use and depression in pregnancy [84], tobacco use and substance abuse correlated with EDS scores. However, our data collection protocol did not allow us to account for the onset of depressive symptoms to further analyze the correlations. It is plausible that depressed or stressed women are likely to smoke tobacco or use marijuana as a coping mechanism [85,86]. It is also plausible that there is no causal relationship between substance use or smoking and depression, but rather the behavior is an expression of one’s attitude or emotion [87]. Lazarus and Folkman [88] described stress as being physiologic and psychologic, both having the opportunity to lead to problematic health outcomes. In the context of this study, we believe stressors lead to physiologic and psychologic depressive responses without regard to co-occurring behaviors such as smoking or marijuana use.

Based on this study, there is a high risk of perinatal depression among African American (27%) compared to the national average of all perinatal women regardless of race (8.5%). In our study of only African American women, those with lower income/poverty ratio at the federal poverty threshold, had disproportionate rates of depression (63.5%) compared to those with higher income or private insurance (17.3%). This is markedly higher than other studies, which have reported elevated perinatal major depression rates of 19–28% [41,42,89,90] also in low-income African American women.

Our outcome measure, EDS score, is a measure of depression *risk*, not a diagnostic measure. It is primarily used by health professionals and researchers to objectively identify mothers suffering from stress that may be interfering with normal activities or enjoyment of life. Clinically, the EDS cutoff used for referral or to warrant repeat testing remains variable [91,92,93]. In light of this uncertainty, we took a careful analytic approach to identify an appropriate value specific for our sample. Surprisingly, our analysis demonstrates that using a standard cutoff of ≥10 is optimal for decision making in regard to depression risk for urban African American pregnant women. Using a cutoff of ≥10 is likely to accurately predict depression in most women but some suggest repeat testing as frequent as two weeks later regardless of the cutoff score [94], especially among populations where cultural variations may exist in the expression of depressive symptoms [95]. Tandon et al. [42] collected depression measures and performed subsequent sensitivities on the EDS. Similar to this study, they recommended an optimal cutoff score for major and minor depression of ≥10 in this population. In general, African Americans are less likely to be diagnosed with depression or treated for depressive symptoms compared to White patients [50], and across subgroups, African Americans are more likely to have persistent mental disorders after becoming ill [51]. At a minimum, providers should be alerted when EDS scores fall between 7–10 since patients could be developing depression and a trustworthy rapport can positively impact patient compliance and follow-up [96,97]. The need for replication and consistent results is suggested especially in light of the findings being preliminary and without sample size calculations. Because of limited research and slow PBDE degradation, it is unclear whether PBDE accumulation has other long-term implications for pregnant women. Additional studies are needed to address population health impacts resulting from PBDE bioaccumulation since the chemicals are likely to be present in the environment for many years ahead.

### 4.1. Limitations

There are notable limitations to this study. Since our study relied on maternal self-report to document tobacco use, substance abuse, and depressive symptoms, it is possible that recall bias interfered with our results. The study sample is based on convenience sampling from two metropolitan prenatal clinics. While it may represent most African American pregnant women living in urban environments, attempts were not made to include hard-to-reach groups. Especially during pregnancy, study participants may have been reluctant to disclose the presence or absence of depressive symptoms or current use of tobacco or illegal drugs. Women report difficulty disclosing perinatal mental illness because of stigmas of inadequacy as a mother or stigmas of treatment in pregnancy [98]. Others refuse to seek help because of poor mental health knowledge or literacy [99]. Self-report of substance abuse is underrepresented resulting from maternal guilt or remorse [100,101]. A cross-sectional study design comprised of one time point limits the ability to cultivate causal relationships. We attempted to compensate for this lack of temporality in part by excluding anyone with a personal or family history of depression. Further longitudinal data is needed to substantiate the findings and more confidently predict conclusions. Another limitation of our analytic approach was the lack of serum lipid data which hindered our ability to represent both the wet weight and lipid adjusted serum PBDE concentrations. We did, however, include BMI, a common surrogate for adiposity, as a covariate to improve depression risk estimates. The debate of whether to include lipid concentrations in the analysis of lipophilic chemicals persists. Variation of PBDE concentrations in humans is common due to multiple exposure routes, elimination differences, and individual factors such as genetics, diet, age, and race [102]. Statistical methods designed to adjust for total lipid serum levels, such as classical standardization, covariate lipid adjustment, and two-stage modeling, have individually been debated for their ability to induce bias when examining health outcomes [103]. Given our study population with carefully controlled inclusion criteria based on race/ethnicity, pregnancy and gestational age, sex, and age, we are positioned to make valuable unadjusted serum PBDE comparisons. Future reports and study designs would be enhanced with greater translation by providing both wet weight and lipid adjusted results. A final key limitation was the inability to determine the major sources of environmental PBDE exposures to this population, which impacts future development of preventive interventions for pregnant African American populations.

### 4.2. Study Implications

Universal screening standards are in place for depression and environmental hazards in pregnancy. For depression screening, the American College of Obstetrics and Gynecology (ACOG) recommends at least one screening in the perinatal period using a validated screening tool in addition to a full emotional well-being exam during the postpartum follow-up visit. At this time, no recommendations are made for specific screening during the antenatal period and this vulnerable screening period may be missed. Our work revealed a high antenatal depression risk of 27% in African American women. At a minimum, this supports additional screening in pregnancy. In terms of identifying environmental risk factors in pregnancy, ACOG has provided a useful committee opinion outlining the effects of environmental exposures on reproductive health and to populations that may be more susceptible to toxicity [104].

In summary, health conditions resulting from environmental hazards are preventable. The ability to modify environments and change human activities that result in pollution is achievable yet have not been universally supported. The intersection between the environment and human health is where public health and medical professionals excel. Whether it is through education, research, advocacy, or providing actual medical care, professions can promote healthy interventions and stimulate change among individuals, communities, and whole populations. Prevention of disease and illness is not only beneficial from an economic standpoint but also in the overall reduction of human suffering.

## 5. Conclusions

Health concerns related to PBDE exposure are becoming more widely acknowledged and will likely continue for years due to slow chemical degradation leading to persistent accumulation in the environment. Products containing PBDEs are common and still used in homes, schools, and businesses across the country. Even those discarded from homes decompose in landfills and recycling centers releasing harmful toxicants into the air, water, and soil. Our analysis revealed BDE-47, -99, and -100 are present in the serum of pregnant African American women and the weighted mixture is associated with high depressive symptoms. The novel associations were found in a relatively small sample size and among a non-occupationally exposed group. Additionally, our sample of African American pregnant women had a high risk of depression compared to the U.S. national average for perinatal women, which includes all racial/ethnic groups. Particularly, African American women living in the Southeastern U.S. may have an increased risk of depression. The cross-sectional methodology of this study can be particularly useful in informing researchers and public health professionals about depressive symptomatology among African American pregnant women and changes in PBDE concentrations over time. Assessing environmental burdens and human toxicant effects, especially mixtures, can inform planning and allocation of health resources for mitigation and prevention. Endocrine disrupting chemicals such as PBDEs are entirely modifiable, and prevention efforts should be given high priority.

## Figures and Tables

**Figure 1 ijerph-18-03614-f001:**
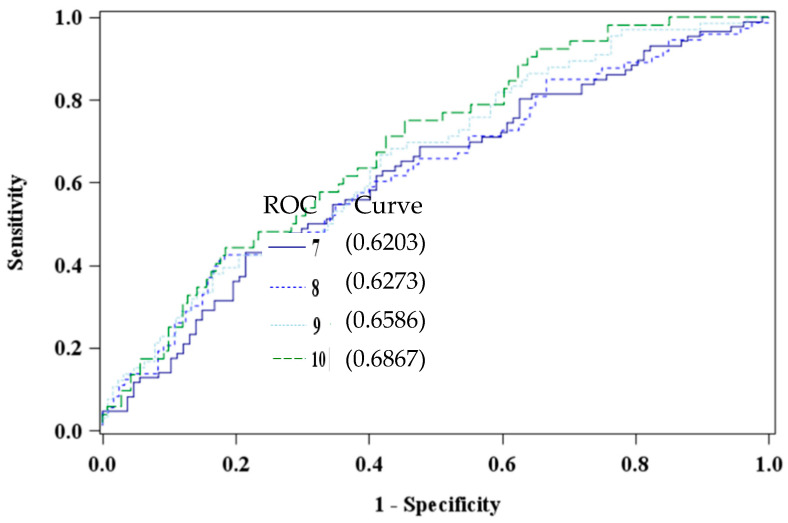
Comparison of Edinburgh Depression Scale Cut Points using PBDE 47 (pg/mL) among African American women at 8–14 weeks gestation (n = 193).

**Figure 2 ijerph-18-03614-f002:**
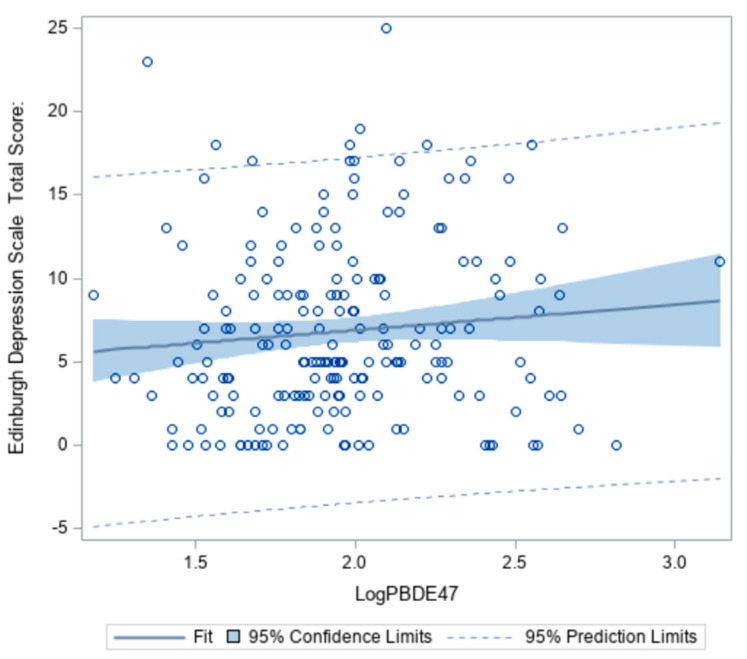
Association between BDE-47 (pg/mL) and Edinburgh Depression Scale Total Score among African American women at 18–14 weeks gestation (n = 193).

**Table 1 ijerph-18-03614-t001:** Demographic characteristics of African American women by high and low or no depressive symptoms (using EDS cutoff ≥10).

Characteristic, n (%)	Total Sample (n = 193)	High Depressive Symptoms (n = 52)	Low or No Depressive Symptoms (n= 141)	*p* Value
Age (mean ± SD)	24.2± 4.4	23.8 ± 3.9	24.4 ± 4.6	0.46
Relationship status (missing = 1)				0.24
Not in relationship	35 (18.1)	7 (13.5)	28 (14.5)	
Relationship not cohabitating	54 (28.0)	17 (32.7)	37 (26.2)	
Relationship cohabitating	103 (53.4)	27 (51.9)	76 (53.9)	
Education				0.3
Some high school	35 (18.1)	12 (23.1)	23 (16.3)	
Graduated high school or GED	65 (33.7)	18 (34.6)	47 (33.3)	
Some college or technical school	63 (32.6)	16 (30.8)	47 (33.3)	
Graduated college	20 (10.4)	6 (11.5)	14 (9.9)	
Some graduate work or degree	10 (5.2)	0 (0.0)	10 (7.1)	
Insurance				
LIM < 100%	96 (49.7)	33 (63.5)	64 (45.4)	0.08
RSM≤ 200%	54 (27.9)	10 (19.2)	43 (30.5)	
Private	45 (23.3)	9 (17.3)	34 (24.1)	

EDS = Edinburgh Depression Scale; GED = general educational development; LIM < 100% = low-income Medicaid insurance for women with poverty/income ratio below the federal poverty level; RSM < 200% = Right from the Start Medicaid insurance for women with poverty/income ratio at or below 200% the federal poverty level. Missing data excluded from analysis.

**Table 2 ijerph-18-03614-t002:** Health-related characteristics of African American women by high and low or no depressive symptoms (using EDS cutoff ≥10).

Characteristic, n (%)	Total Sample (n = 193)	High Depressive Symptoms (n = 52)	Low or No Depressive Symptoms (n = 141)	*p* Value
Body mass index (BMI)				0.03 *
Underweight (<18.5)	8 (4.2)	4 (7.7)	4 (2.8)	
Normal (18.5–< 25)	80 (41.5)	24 (46.2)	56 (39.7)	
Overweight (25–< 30)	39 (20.2)	14 (26.9)	25 (17.7)	
Obese (≥30)	66 (34.2)	10 (19.2)	56 (39.7)	
Number of children				0.63
0		28 (53.9)	65 (46.1)	
1	100 (51.8)	14 (26.9)	44 (31.2)	
2+		10 (19.2)	32 (22.7)	
	58 (30.0)			
	35 (18.1)			
Drinks alcohol in last month	8 (4.2)	4 (50.0)	4 (50.0)	0.14
Ever smoked tobacco	21 (10.9)	14 (26.9)	7 (5.0)	<0.0001 *
Smoked marijuana in last month	33 (17.1)	14 (26.9)	19 (13.5)	0.03 ^*^

EDS = Edinburgh Depression Scale. * Statistically significant at *p* < 0.5.

**Table 3 ijerph-18-03614-t003:** PBDE metabolite serum concentrations (pg/mL) among African American women at 18–14 weeks gestation (n = 193) compared to 2013–2014. NHANES pooled PBDE metabolite data.

Metabolite	GM ± GSD	NHANES GM ± GSD	% Detected in Study Sample
PBDE 47	90.0 ± 2.1	83.1 ± 4.3	100%
PBDE 99	21.2 ± 2.6	16.2 ± 0.9	81%
PBDE 100	13.7± 3.0	18.0 ± 1.0	79%

GM = geometric mean; GSD = geometric standard deviation; Results reflect weighted geometric means and standard deviations from log normalized data without lipid adjustment. NHANES measurements are made from pooled serum samples from non-Hispanic Black women aged 20–39 years.

**Table 4 ijerph-18-03614-t004:** Mean PBDE concentrations (pg/mL) in serum and risk of antenatal depression among African American women at 8–14 weeks gestation (n = 193).

Characteristic	High Depressive Symptoms (n = 52)	Low or No Depressive Symptoms (n = 141)	*p*-Value
**PBDE 47** **PBDE 99** **PBDE 100**	110.9 112.1 104.7	91.9 91.4 94.1	0.04 * 0.02 * 0.24

* Statistical significance at *p* < 0.05.

**Table 5 ijerph-18-03614-t005:** PBDE concentrations and high antenatal depression risk among. African American pregnant women (cutoff ≥ 10).

Metabolite	OR [95% CI]	AOR [95% CI]
**PBDE 47**	2.62 (1.00, 6.87)	4.43 (1.47, 13.40) *
**PBDE 99**	1.39 (0.99, 1.93)	1.58 (1.08, 3.00) *
**PBDE 100**	1.17 (0.88, 1.58)	1.23 (0.90, 1.68)

Models were adjusted for relationship status, insurance type, and BMI; CI = confidence interval; * Statistical significance at *p* < 0.05.

**Table 6 ijerph-18-03614-t006:** Weights of measured PBDEs in association with high antenatal depression risk among African American pregnant women using Weighted Quantile Sum (WQS) regression.

Metabolite	Weights
**PBDE 47** **PBDE 99** **PBDE 100**	0.71 0.00000000817 0.29
**WQS**	AOR [95% CI] 2.93 (1.18, 7.82) *

Models were adjusted for relationship status, insurance type, and BMI; CI = confidence interval; * Statistical significance at *p* = 0.02.

## Data Availability

Data and associated documentation may be made available only under a data-sharing agreement that provides for: (1) a commitment to using the data only for research purposes and not to identify any individual participant; (2) a commitment to securing the data using appropriate computer technology; and (3) a commitment to destroying or returning the data after analyses are completed.

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
