# Peer review of "Polybrominated Diphenyl Ether Serum Concentrations and Depressive Symptomatology in Pregnant African American Women"

_ijerph, 2021, doi:10.3390/ijerph18073614_

Round 1

Reviewer 1 Report

Mutic et al explored the associations between serum PBDE concentrations and depressive symptomatology in pregnant African American women. The present study was designed well and interpretation of the obtained data was also logical. The only thing that I concerned is that some statistical results are not very convincing. For example, the authors stated that levels of BDE -47 and -99 were significantly different between low and high depression groups (p=0.04 and p=0.02, respectively). However, from table 4, we can see that mean concentrations (±SD) of PBDE-47 in low and high depression groups were 113.7±105.8 and 155.3±198.3, respectively. And mean concentrations (±SD) for PBDE-99 were 91.4±342.9 and 112.1±342.9, respectively, in low and high depression groups. These data, roughly, doesn’t match well with their reported p values of 0.04 and 0.02 in the present study. This is just one case, and there is some other places with results looking weird. Please double check your statistical analysis process. Besides, another imperfection is that the reported PBDE concentrations in the present study was not corrected with lipid content, which may bias the observed results. The authors should give sufficient reasons that why lipid was not measured to adjusted the PBDEs concentrations.

Reviewer 2 Report

Journal: IJERPH

Title: Polybrominated Diphenyl Ether Serum Concentrations and Depressive Symptomatology in Pregnant African American Women

Manuscript Number: IJERPH-1042464

Keywords:

 polybrominated diphenyl ether; antenatal depression; endocrine disrupting chemical; 27 neuroendocrine.

Thank you for the opportunity of reading and reviewing this interesting study.

Abstract:

The assumption the authors make in respect to interventions do not fall within the aim of this study. I would delete that sentence and substitute for a more appropriate one.

Please include the measure used to assess depressive symptoms.

Introduction

From line 30 to 41 there is a lot of information, but only a single reference. Please, add some more references supporting the information the authors include in those lines.

Line 48: “two studies” should not be underlined.

Line 49: “and” is in italics. If this is a typo, please amend.

Line 50: Instead of “pregnant adults”, it is more appropriate to use “pregnant women over 18 years old”

Line 50: “Relationship” is more common for social relationship. In this context, the term “association” would be better.

Is there any connection between the thyroid function and depressive symptoms? Or is the effect of PBDE on depression symptoms more associated with a brain damage? In both cases, please clarify in the introduction.

What is EDS? Is it the Edinburgh Depression Scale? It should be explained the first time they use the acronym.

Why is none underlined?

Methods

Please, clarify why the authors decided to use a measure that was first designed for a postpartum use

Results

The way the authors divided their sample into Group 1 (High depressive symptoms) and Group 2 (Low depressive symptoms) is appropriate. Nevertheless, it is not appropriate to compare these 2 groups when the amount of participants is, for example, n = 3 in the married group and n = 23 in the married group. These 2 groups are not proportional and sample size is an issue here. Please delete the married group from table 1. Instead use a more appropriate approach using your data.

Please include in Table 1 and Table 2 the statistical used to compare groups. The same applies to Table 3 and 4.

Figure 2 is a good one. I would move the legend related to the statistics to one side so it can be clearly read.

Discussion

Please include some clinical and research implications of this study.

Reviewer 3 Report

abstract must cleary exsplained

introduction is too long

section of result is lacking

Reviewer 4 Report

Thank you very much for allowing me to review the article "Polybrominated Diphenyl Ether Serum Concentrations and Depressive Symptomatology in Pregnant African American Women" (ijerph-1042464).

The aim of this study is to investigate   whether   PBDE   exposure   in   pregnant   AA   women   is   associated   with   depressive symptomatology during pregnancy.

Comments:

The objective should be rewritten indicating more precisely the objective of the work. The objective should be at the end of the introduction and the hypotheses before the objective.

Introduction: The basis for which AA is considered to be more exposed to PBDEs should be explained.

Material and methods Line 97, please state the number "larger longitudinal pregnancy and birth cohort". A flow chart of the mothers participating in the study from which their participation was proposed should be included. This diagram will allow assessing the representativeness of the sample studied.

Lines 111 and 112, indicate how the variables studied have been collected, in what categories and based on what criteria.

2.3.

Clinical Data It would be interesting to know if the causes of depression have been assessed, in case there is depression of known cause. This aspect is a potential confounding bias in the study. It should also be valued in the discussion.

Please explain "Insurance, Medicaid <100%, Medicaid≤ 200% and Private"

Results

Tables and figures provide a better understanding of the work. But some methodological aspects should be in material and methods not in results.

In the conclusions, it should be assessed that since there is no sample size calculation for the study, it should be assessed as a preliminary study, that more studies are necessary to confirm the results. So your conclusions should be adjusted to your results. In addition to being cautious that a cross-sectional design, such as this one, allows establishing associations but does not allow the establishment of causality.

Round 2

Reviewer 4 Report

After reviewing the authors' responses to my comments and reviewing the new version of the article "Polybrominated Diphenyl Ether Serum Concentrations and Depressive Symptomatology in Pregnant African American Women" (ijerph-1042464).

 I have verified that the authors have incorporated the suggestions made. I also want to indicate that I fully agree with the comments made by the other reviewers.

This work allows us to identify that PBDE exposure may have significant clinical implications in postpartum depression.

Author Response

Thank you for your review and suggestions to better this work.